# Extracellular Vesicles as Potential Bladder Cancer Biomarkers: Take It or Leave It?

**DOI:** 10.3390/ijms24076757

**Published:** 2023-04-04

**Authors:** Ana Teixeira-Marques, Catarina Lourenço, Miguel Carlos Oliveira, Rui Henrique, Carmen Jerónimo

**Affiliations:** 1Cancer Biology and Epigenetics Group, Research Center of IPO Porto (CI-IPOP)/RISE@CI-IPOP (Health Research Network), Portuguese Oncology Institute of Porto (IPO Porto)/Porto Comprehensive Cancer Center Raquel Seruca (Porto.CCC Raquel Seruca), 4200-072 Porto, Portugal; 2i3S-Instituto de Investigação e Inovação em Saúde, Universidade do Porto, 4200-135 Porto, Portugal; 3INEB—Instituto Nacional de Engenharia Biomédica, Universidade do Porto, 4200-135 Porto, Portugal; 4Doctoral Programme in Biomedical Sciences, School Medicine and Biomedical Sciences, University of Porto (ICBAS-UP), 4050-313 Porto, Portugal; 5Department of Pathology, Portuguese Oncology Institute of Porto (IPOPorto), 4200-072 Porto, Portugal; 6Department of Pathology and Molecular Immunology, School of Medicine & Biomedical Sciences, University of Porto (ICBAS-UP), 4050-313 Porto, Portugal

**Keywords:** bladder cancer, liquid biopsies, extracellular vesicles, biomarkers, lncRNA, protein, miRNA

## Abstract

Bladder cancer (BC) is the 10th most frequently diagnosed cancer worldwide. Although urine cytology and cystoscopy are current standards for BC diagnosis, both have limited sensitivity to detect low-grade and small tumors. Moreover, effective prognostic biomarkers are lacking. Extracellular vesicles (EVs) are lipidic particles that contain nucleic acids, proteins, and metabolites, which are released by cells into the extracellular space, being crucial effectors in intercellular communication. These particles have emerged as potential tools carrying biomarkers for either diagnosis or prognosis in liquid biopsies namely urine, plasma, and serum. Herein, we review the potential of liquid biopsies EVs’ cargo as BC diagnosis and prognosis biomarkers. Additionally, we address the emerging advantages and downsides of using EVs within this framework.

## 1. Introduction

### 1.1. Bladder Cancer

#### 1.1.1. Epidemiology and Biology

Bladder cancer (BC) ranks as the 10th most common malignancy worldwide, showing a high incidence in regions with a high human development index, such as Europe and North America, in which it constitutes the fourth most common cancer in men and ninth most common in women [1,2]. Because life expectancy has risen globally and BC mostly afflicts the elderly, the incidence has increased over the last 20 years [3]. Furthermore, given such demographic trends, its global health burden is likely to further grow in the near future [1]. BC diagnosis depends on the transurethral resection of the bladder tumor (TURBT), enabling the complete removal of visible lesions under direct cystoscopic examination [4]. This technique may be complemented with urine cytology, often used as an ancillary procedure for BC detection. 

Most BCs originate in the urothelium, the epithelial tissue that lines the lumen of bladder and urinary organs, making urothelial carcinoma the most common type of BC (90% of all cases) [5]. The disease may be further stratified based on the tumor’s ability to invade the muscle layer. Non-muscle-invasive BC (NMIBC) comprises about 70% of newly diagnosed tumors, while the remaining 30% are muscle-invasive BC (MIBC). Importantly, BC has long been recognized as a heterogenous and complex disease, presenting multiple features that challenge clinicians and researchers. 

#### 1.1.2. Current Hurdles and Disease Management

Concerning NMIBC, frequent recurrence and progression (up to 50–70% and 10–30%, respectively) constitute the major clinical problems [6,7,8]. Because patients enduring relapse and/or progression cannot be prospectively identified, rigorous and, in many cases long-term, surveillance is required [8,9,10,11,12]. Indeed, currently available patient risk stratification parameters, solely reliant on clinicopathological variables, are imperfect and incapable of portraying the true heterogeneity and complexity of BC [8,9,13]. Consequently, BC is the costliest cancer to treat on a per-patient basis, particularly driven by periodic and invasive cystoscopies, leading to a significant financial burden to healthcare systems [8,14,15,16,17,18]. Furthermore, there is considerable patient morbidity, as cystoscopies frequently originate anxiety, pain, hematuria, and even urinary tract infections [19,20]. Urine cytology, although noninvasive and reliable (90–95% specificity), shows low sensitivity (30–50%) for BC detection, and, consequently, cystoscopy cannot be spared to check for recurrences during patients’ follow-up [21,22,23]. Whereas alternative urine tests have been developed, such as bladder tumor antigen (BTA) and Nuclear Matrix Protein 22 (NMP22) assessment, showing higher sensitivity (50–70%), the specificity remains suboptimal (60–90%); therefore, these are not recommended at present since they do not obviate the need for cystoscopy [21,23,24].

The identification of novel, accurate, cost-effective, and noninvasive cancer biomarkers has, thus, become a fundamental goal of research on NMIBC [17]. In addition to addressing the aforementioned shortcomings, the implementation of novel biomarkers in clinical practice might also provide improved risk stratification, identifying which patients might benefit from further therapeutic interventions, as well as those with low-risk disease who should be spared excessive interventions. Overall, these should allow for the design of more effective follow-up strategies, enabling the earlier detection of disease recurrence and progression, and simultaneously reducing morbidity due to frequent monitoring. 

MIBC lies on the opposite side of the spectrum. This is an aggressively invasive and rapidly metastatic disease, carrying a high mortality risk (40–60% 5-year survival) [25]. The major clinical problem is treatment failure due to inaccurate patient selection, prompting unnecessary costs to the patient and the healthcare system [9]. This may be attributed to the lack of adequate tools for patient selection and, consequently, treatment is mostly offered as “one size fits all” [9,26]. Similar to NMIBC, the identification of accurate and predictive biomarkers for therapy response would improve patient outcome and avoid ineffective treatment in probable nonresponders [9]. 

### 1.2. Extracellular Vesicles (EVs): A New Source of BC Biomarkers in Liquid Biopsy

Liquid biopsies have been gaining increasing attention in recent years. They encompass the minimally or noninvasive sampling of biological fluids, such as plasma, serum, or urine, and their contents are a potential source of biomarkers. Importantly, they are a minimally or noninvasive, fast, and affordable means of acquiring relevant clinical information, enabling earlier diagnosis as well as real-time disease monitoring, and granting a personalized snapshot of disease evolution—a core prerequisite of precision medicine [27,28,29,30,31,32]. Compared to tumor tissue samples, the gold standard for diagnosis and prognostication, liquid biopsies may be performed in a serial manner, providing a better understanding of disease evolution over time, more accurately reflecting the diversity of tumor subclones, and providing a wider and more complete picture of the tumor, an attribute of particular relevance in heterogenous cancers such as BC [29,31]. 

#### 1.2.1. EVs’ Biogenesis

EVs are a heterogenous population of lipid enclosed structures abundantly present in body fluids [33]. According to their mechanism of assembly, they may be classified into three main categories: exosomes (30–150 nm, formed by the fusion of multivesicular bodies with the plasma membrane); microvesicles (100–1000 nm, generated by direct budding from the cell membrane); and apoptotic bodies (50–5000 nm, released during programmed cell death) [34,35,36]. EVs found in liquid biopsies likely represent a mixture of vesicles originating from all three biogenesis pathways, with considerable size and density overlap among them [36]. As currently available purification methods are incapable of fully discriminating according to their biogenesis, the use of the generic term EV is recommended by the Minimal Information for Studies of Extracellular Vesicles (MISEV) guidelines [37]. 

#### 1.2.2. EVs’ Physiological and Pathological Role

EVs have emerged in recent years as key mediators of paracrine and endocrine intercellular communication in both physiological and pathological processes. They serve as vehicles for the transfer and delivery of proteins, lipids, DNA, RNA, and metabolites to recipient cells, shielded from degradation by the lipid bilayer membrane [28,34,36,38,39]. This mechanism is acknowledged to play a leading role in tumorigenesis. Specifically, through the transfer of protumoral cargo, EVs may stimulate cell proliferation, promote angiogenesis, induce drug resistance, modulate the microenvironment, and support the establishment of premetastatic niches [28,36,40,41,42,43,44]. For instance, EVs from BC cells internalized by macrophages promote their polarization into protumoral macrophages, enhancing the release of immunosuppressive cytokines, which facilitates tumor progression [45]. Additionally, the transfer of EVs’ lncRNAs and miRNAs from cancer-associated fibroblasts to BC cells showed chemotherapy resistance modulation [46,47]. Moreover, proteins derived from EVs of BC cells increased angiogenesis and the migration of BC cells as well as endothelial cells, also facilitating cancer progression [48,49]. These studies provide evidence of EVs and its cargo’s protumoral influence in BC.

Considering the aforementioned mentioned challenges in the management of BC patients, especially the lack of accurate biomarkers for early detection and prognostication, we explore in this review the published literature on the potential of BC-EV-derived biomarkers as a noninvasive tool to assist in the clinical management of BC patients, as well as the limitations of such studies.

## 2. Methods

For this review, a PubMed database search was performed on 15 January 2023, using the query: (Extracellular vesicles OR Exosomes OR Microparticle) AND (Bladder Cancer OR Bladder Neoplasm) AND (Biomarkers OR Transcriptome OR Molecular Markers) AND (Blood OR Plasma OR Serum OR Urine), with no time interval restraints. Only original records published in English were considered (reviews were excluded). Records were first screened through critical abstract evaluation, followed by full-text read-outs and the selection of those providing meaningful information regarding the topic to be included in the final analysis. A flow chart summarizing the methodology is provided in Figure 1. Information regarding the biomarkers depicted in the different studies is illustrated in Table 1, Table 2, Table 3 and Table 4, with Figure 2 summarizing the candidate biomarkers’ distribution among different biofluids. Moreover, Figure 3 presents an overview regarding the isolation methods used in the review studies. Regarding the tables, the patient cohorts’ designation given by the authors was, when possible, maintained, regardless of its size or goal. If the cohort’s name was not defined by the authors, we considered cohort 1, 2, or 3 depending on the number of independent cohorts used in the study. Furthermore, the term healthy control (HC) comprises the denominations “healthy control”, “healthy”, “control”, or “healthy donor” used by the original authors. The designation “benign lesions” was used whenever a patient had a lesion suspected to be cancerous that, upon initial assessment, turned out to be a nonmalignant condition. Benign urological diseases comprise benign pathologies such as urinary lithiasis, benign prostate hyperplasia, obstructive uropathy, and nonspecified benign conditions of urologic origin. Only the best outcomes are shown, except when multiple markers and/or panels are worth mentioning owing to different advantages/benefits in performance measures. Finally, regardless of the denomination used by the authors in the original manuscripts, the term EV was used in this review.

## 3. EVs in BC

### 3.1. miRNA Biomarkers in BC

After performing a miRNA array and qRT-PCR analysis in urinary EVs (uEVs), Andreu et al. highlighted miR-375 and miR-146a as diagnostic markers of high-grade and low-grade BC, respectively [55]. Moreover, Matsukazi et al. identified miR-21-5p as a highly valuable biomarker for BC diagnosis (sensitivity, 75.0%; specificity, 95.8%), also disclosing higher levels in uEVs from BC patients with negative urine cytology [54]. El-Shal et al. chose up-regulated EV-derived miRNAs previously reported in the literature to develop a diagnostic panel for BC, with high specificity (87.8%) and sensitivity (88.2%) for detecting BC using combined levels of miR-96-5p and miR-183-5p, which also correlated with clinicopathological features [52].

Combining high-throughput sequencing and miRNA BC tissue levels from the TCGA database, uEV-derived miRNA candidates were validated with qRT-PCR in an independent cohort, resulting in the identification of both miR-93-5p and miR-516a-5p as potential BC diagnostic biomarkers. Interestingly, miR-93-5p also discriminated MIBC from NMIBC [51]. Using the next-generation sequencing of matched urine and serum-EV-derived miRNA from BC patients pre- and postsurgery, Strømme et al. identified two miRNAs in uEVs (miR-451a and miR-486-5p) that were significantly up-regulated in presurgery samples from T1 patients compared to postsurgery check-up samples. Moreover, no differential miRNA levels were found in the serum of these patients. This study suggests that uEV-derived miR-451a and miR-486-5p are potential biomarkers of recurrence-free survival in T1 BC [50]. Baumghart et al. sought to refine MIBC patient selection for radical surgical treatment. Thus, uEVs were isolated and the results were compared with those of formalin-fixed paraffin-embedded (FFPE) tumor tissues. MiR-146b-5p and miR-155-5p were up-regulated in MIBC patients compared to NMIBC, indicating that they discriminate MIBC from NMIBC [53]. 

Concerning studies exploring plasma, Yin et al. showed that miR-663b levels assessed with qRT-PCR were elevated in BC patients [57]. Additionally, Yan et al. isolated EVs with size exclusion chromatography (SEC) and demonstrated that miR-4644 was up-regulated in BC compared to HC [56]. 

### 3.2. lncRNA Biomarkers in BC

Contrarily to the studies on EVs’ proteins and miRNAs, that usually carried out cargo profiling, EV-derived lncRNA studies focused mostly on evaluating the potential of preselected candidates. For instance, Zhan et al. isolated uEVs using the Exosomal RNA Isolation Kit (Norgen), and by performing RT-qPCR, they assessed the levels of eight lncRNAs in a training set. A final panel for BC detection comprising the lncRNAs MALAT1, PCAT-1, and SPRY4-IT1 showed a superior AUC (0.813) compared to urine cytology (0.619) in a validation set. Furthermore, PCAT-1 and MALAT1 levels were associated with shorter recurrence-free survival in NMIBC patients [60]. Using a commercial RT-qPCR precipitation, Abbastabar et al. found that T1 and T2 BC patients displayed higher ANRIL and PCAT-1 levels in uEVs compared to HC, achieving 46.67 % sensitivity and 87.5% specificity for ANRIL and 43.3% sensitivity and 87.5% specificity for PCAT-1 [59]. In another study, using sequencing for RNA profiling, Chen et al. found that uEVs’ TERC levels were higher in BC patients than in HC, with a diagnostic performance of 78.65% sensitivity and 77.78% specificity, which is considerably higher than that of the NMP22 (FDA-approved) test and urine cytology. Additionally, the TERC levels discriminated low-grade from high-grade disease [58]. 

Zhang et al. selected and quantified 11 candidates in a training set of BC and HC serum samples to predict and detect BC recurrence. Among those, three lncRNAs were up-regulated in patients compared to HC. Subsequently, in a validation set, the three-lncRNA panel (PCAT-1, UBC1, and SNHG16) detected BC with 80% sensitivity and 75% specificity, outperforming urine cytology. UBC1 and SNHG16 were also up-regulated in MIBC vs. NMIBC, thus associating with deep bladder wall invasion. Moreover, UBC1 was also suggested to serve as a prognostic marker, since higher levels associated with poor recurrence-free survival in NMIBC [61]. Wang et al. explored serum-EV-derived H19 as a BC biomarker. After ensuring that measured H19 derived only from inside EVs, the authors observed that H19 levels were increased in BC patients compared to HC and benign disease, further correlating with tumor stage. Moreover, the postoperative samples presented decreased lncRNA levels compared to the preoperative samples. Interestingly, BC patients with higher H19 levels endured shorter overall survival [62]. In another study, PTENP1 was found to be decreased in plasma EV from BC patients and paired BC tissues. Biologically, PTENP1 expression increased cell apoptosis and reduced invasion and migration [63]. Finally, released lncRNAs may induce tumor growth and progression during hypoxia. Xue et al. reported that UCA1 promoted BC cell proliferation, migration, and invasion. After assessing the relevance of this lncRNA in cell lines, the authors confirmed that UCA1 was elevated in the serum of BC patients, compared to HC [64].

### 3.3. Protein Biomarkers in BC

Urine has been the fluid of choice for assessing free proteins as biomarkers for BC, and the study of uEV proteins has followed the same trend. Proteomic analysis using liquid chromatography–tandem mass spectrometry (LC-MS) demonstrated an enrichment of several proteins in the uEVs of BC patients compared to HC [69,70,71,72]. However, only Chen et al. confirmed the potential of the tumor-associated calcium signal transducer 2 (TACSTD2) in uEVs for BC diagnosis [71]. Furthermore, Tomiyama et al. used density gradient ultracentrifugation (DUC) to isolate uEVs and carried out a combined proteomic analysis of uEVs and EVs derived from tissue exudate. After performing tandem mass tag (TMT)-LC-MS/MS analysis, 22 membrane proteins were selected as BC candidate biomarkers for validation, using selected reaction monitoring/multiple reaction monitoring (SRM/MRM) analysis on an independent cohort of 70 individuals. Heat-shock protein 90, syndecan-1, and myristoylated alanine-rich C-kinase substrate (MARCKS) were validated as significantly up-regulated in BC patients [68]. 

Surprisingly, and despite plasma being widely used for biomarker research, there are no studies on BC-EV-derived proteins in this biofluid, to the best of our knowledge.

### 3.4. Other Molecules as BC Biomarkers

Yazarlou et al. analyzed the MAGE-B4 profile in uEVs, highlighting its potential for BC diagnosis. Its expression, however, was higher in patients with benign prostate hyperplasia (BPH) than in BC patients [74]. Moreover, Amuran et al., after uEV isolation using the Norgen kit, combined EV-derived miR-139-5p, miR-136-3p, miR-19b1-5p, and miR-210-3p with the urinary proteins Ape1/Ref1, BLCA4, CRK, and VIM into a panel able to discriminate BC (especially low-risk) patients from HC with 93.3% sensitivity and 95.5% specificity and 80.0% sensitivity and 86.4% specificity, respectively [77]. Lastly, using high-throughput RNA-Seq in uEVs, Huang et al. established and validated a panel combining the mRNAs KLHDC7B, CASP14, and PRSS1 and the lncRNAs MIR205HG and GAS5, which discriminated BC patients from HCs with excellent performance (88.5% sensitivity and 83.3% specificity). Additionally, RNA levels were associated with tumor stage and grade [76]. 

## 4. Discussion

Presently, cytology is the only test implemented for assisting in BC patient management, used in complementarity with cystoscopy [9]. The FDA-approved urine-sediment-based tests for BC diagnosis and follow-up, mostly assessing proteins, metabolites, DNA, or mRNA, have not gained wide acceptance in clinical routine due to its low sensitivity for detecting early disease and its limited reproducibility [81]. Thus, novel molecular biomarkers are required to fill this gap. In this scenario, EVs have shown potential as a source of biomarkers, with interest growing exponentially over the years. The clinical drive for studying EVs lies in their critical role in a comprehensive range of pathological processes in several cancers [38]. 

### 4.1. The Potential of EVs as BC Biomarkers

EVs may be found in almost all biofluids, each containing different information that may potentially answer different questions concerning BC management. Whereas urine, collected in a noninvasive manner, may be more informative for detecting early-stage BC, plasma provides the advantage of being less influenced by bladder inflammation and may even allow for detecting premetastatic signals. Due to EVs’ presence in these biofluids and their potential as minimally invasive biomarkers, the role of EVs as BC biomarkers has been extensively explored (Figure 2).

Thus far, most studies have focused on BC diagnosis and most disclose biomarker performance parameters, generally reporting higher sensitivity than cytology for BC detection (Table 1 and Table 2), underscoring the promising value of EV-derived biomarkers as diagnostic tools [58,60-68,71,74,76,77]. Moreover, specific serum lncRNAs and miRNAs’ levels have been associated with recurrence-free survival and overall survival, suggesting that EVs may be useful for prognostication [50,60,62]. Additionally, EV-derived biomolecules were found to be differentially regulated according to clinicopathological features such as the tumor grade and the level of bladder wall invasion, unveiling an interconnection between biomarker levels and disease aggressiveness [52,53,55,56,70,76]. Importantly, two clinical trials using EVs as BC biomarkers are ongoing: *miR Sentinel BC* uses uEV-derived miRNAs for BC detection and monitoring in patients with hematuria (NCT04155359), whereas uEV lncRNAs are used at diagnosis to stratify patients according to lymph node metastatic status (NCT05270174).

### 4.2. Challenges and Drawbacks in BC-Derived EV Research

Although there is a plethora of studies proposing EV-derived biomarkers and elucidating their potential, very few have made their way into clinical trials due to preanalytical issues and a lack of standardized reporting. For instance, the published studies focusing on the discovery of EV-based biomarkers in BC report different candidates. Nevertheless, PCAT-1 and PRMT5 were shown to be present both in serum and urine [59,60,61,80]. Additionally, uEV-derived PCAT-1, MALAT-1, and EPS8L2 are represented in different studies [59,60,61,69,72,78]. The lack of overlapping may be explained by differences in experimental design, such as the preprocessing conditions or the EV isolation method, as well as the dissimilar composition and size of the patient cohorts [82,83,84].

#### 4.2.1. Study Design Constraints

Some of these studies use small cohorts of patients and controls. Thus, after unveiling and testing potential candidates, validation in larger studies from different institutions is needed to disclose the real clinical value of these biomarkers. Importantly, the proportion of MIBC and NMIBC stated in most reports does not mirror the real-world patient distribution, which may bias the biomarker’s performance results. 

#### 4.2.2. Limitations of EVs’ Isolation

Another major shortcoming is the type of EV isolation method. Despite advances in the EV field, the challenge of efficient EV isolation is far from being overcome, particularly in biofluids, owing to their complexity and variability [85]. Although several different methods have been developed, such as differential ultracentrifugation (UC) and polymer precipitation, method standardization is lacking, and each may be performed in a variety of ways [38]. 

UC stands as the most widely used method for uEV isolation for BC biomarker discovery (Figure 3). Although UC provides EV samples with adequate recovery, most protocols remain time-consuming, on top of requiring a large volume of each sample, limiting clinical application. Moreover, commercial kits, mainly ExoQuick, are often used in plasma and serum EV studies (Figure 3). These are usually costly and provide EV samples of low purity. Thus, the variety and complexity of these protocols hinder a comprehensive profiling of EV cargo. Thus, developing a cost-effective protocol that requires lower runtimes and volume inputs, more amenable for clinical use, is of utmost importance.

### 4.3. Constraints in EVs’ Cargo Analysis

Full EV characterization and/or the presence of contaminants is often not reported in several studies. Consequently, concerns about the intravesicular origin of the identified candidates cannot be disregarded. Furthermore, treatments such as RNAse and proteinase, that may affect biomarker results, are often not performed. For instance, when targeting a nucleic acid, authors should consider DNAse and RNAse treatments before extraction to assure that the biomarker of interest is, in fact, EV-derived. This is particularly relevant concerning EV-derived-protein studies. Indeed, some authors sought to perform protease treatments before the analysis of EV-derived proteins to ensure that the cargo originated from EVs. However, treatment may partially damage EV membranes and, consequently, degrade EVs’ internal molecules [86].

Finally, whereas EV-derived-protein analysis methods, such as ELISA, have the advantage of not requiring normalization, studies on RNA used RT-qPCR to evaluate and quantitate RNA levels in a relative manner (Table 1 and Table 2). Nonetheless, housekeeping biomolecules for EV cargo normalization are not consensual, limiting unbiased assessment, and further leading to non-reproducible results [87]. Thus, it is of vital importance to uncover and validate housekeeping molecules within EVs and/or apply techniques that evaluate levels in an absolute manner, such as droplet digital PCR. 

### 4.4. Future Perspectives

Considering the drawbacks of available biomarkers for BC detection and prognostication, novel biomarkers are urgently needed. Indeed, EVs’ derived cargo has been shown to be powerful tools as BC biomarkers. Additionally, several reports have already suggested EVs and their cargo as potential cancer vaccines in different cancer models [88]. Because therapeutic options are very limited in BC, it is of utmost importance to unveil alternative therapies, eventually using EVs as a therapeutic option. Indeed, a clinical trial is ongoing focusing on chimeric EVs vaccine administration to treat patients with recurrent or metastatic BC (NCT05559177).

Although promising, EV-related methodological hurdles are considerable. In the next few years, research must focus on addressing the technical shortcomings of EVs isolation, not only by developing and standardizing techniques that might be easily (and reproducibly) implemented in clinical practice, but also by using methods that allow for accurate EV cargo quantification (e.g., ddPCR or the large-scale validation of housekeeping molecules for RT-qPCR analysis). The clinical validity issues must be also solved by increasing patient cohorts as well as by performing multicenter validation to ascertain the real value of EVs as BC biomarkers. 

## 5. Conclusions

EV-derived miRNAs, lncRNAs, mRNAs, and proteins may serve as biomarkers for both BC diagnosis and prognostication. Not only might they improve patient care through a more precise and minimally invasive strategy, but they might aid in overcoming contemporary challenges in the field. Importantly, technical issues still hamper their use in a clinical setting. Nonetheless, research on EVs is advancing at a fast pace, showing their great potential as a source of biomarkers, which further emphasizes the value that EV-derived molecules may have in the clinical management of BC patients.

## Figures and Tables

**Figure 1 ijms-24-06757-f001:**
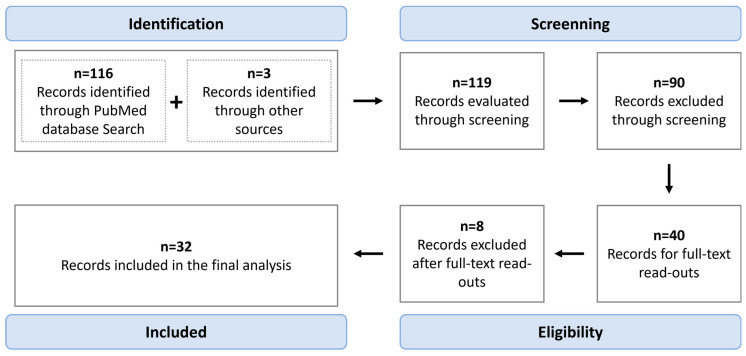
Flow diagram summarizing the methodology used in this review.

**Figure 2 ijms-24-06757-f002:**
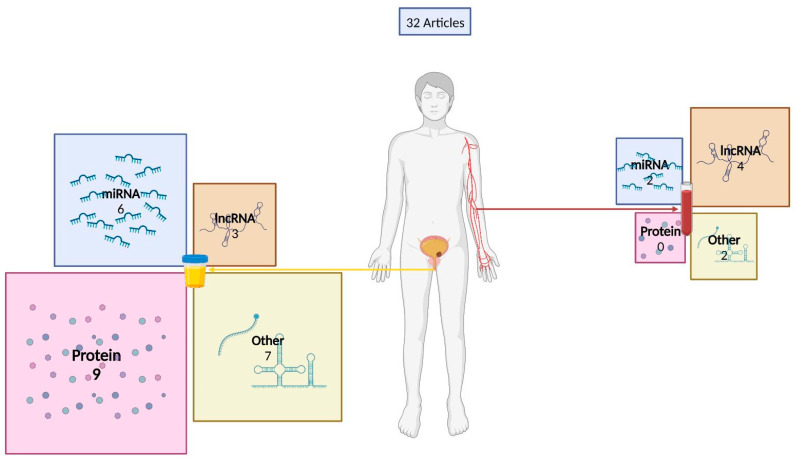
Distribution of extracellular-vesicle-derived bladder cancer biomarkers within urine and plasma/serum biofluids. Numbers represent the number of BC-EV biomarker studies. Created with BioRender.com.

**Figure 3 ijms-24-06757-f003:**
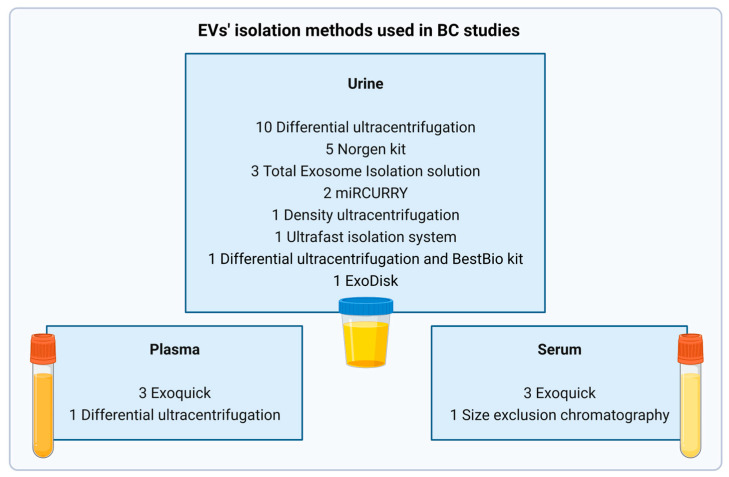
Summary of extracellular vesicle isolation methods that were used in BC articles, concerning urine, plasma, and serum. Numbers represent the number of BC-EV biomarker studies. Abbreviations: BC—Bladder cancer; EVs—extracellular vesicles. Created with BioRender.com.

**Table 1 ijms-24-06757-t001:** miRNAs with potential in the management of bladder cancer patients.

**Study**	**Cohort**	**miRNA**	**Levels**	**SN (%)**	**SP (%)**	**AUC**	**Clinical Significance**
**Urine**							
Strømme et al. 2021 [50]	41 NMIBCs and 15 HCs	miR-486-5p miR-451a	↑	-	-	-	Prognosis: pre- vs. postsurgery
Lin et al. 2021 [51]	Discovery: 6 NMIBCs, 6 MIBC, and 4 HCs	miR-516a-5p	↑	72.9	89.9	0.79	Diagnosis: BC vs. HC
		miR-93-5p		74.1	90.2	0.838	
	Validation: 32 NMIBCs, 21 MIBCs, and 51 HCs			90.5	60.6	0.769	Diagnosis: MIBC vs. NMIBC
El-Shal et al. 2021 [52]	22 NMIBCs, 29 MIBCs, 21 benign lesions, and 28 HCs	miR-96-5p	↑	80.4	91.8	0.85	Diagnosis: BC vs. HC
		miR-183-5p		78.4	81.6	0.83	
		Combined panel: miR-96-5p and miR-183-5p		88.2	87.8	0.878	
Baumgart et al. 2019 [53]	Validation: 17 NMIBCs and 20 MIBCs	miR-146b-5p	↑	-	-	-	Diagnosis: MIBC vs. NMIBC
Matsuzaki et al. 2017 [54]	6 BC and 3 HCs	miR-21-5p	↑	75.0	95.8	0.90	Diagnosis: BC vs. HC
	Validation: 18 NMIBCs, 18 MIBCs, and 24 HCs						
Andreu et al. 2016 [55]	Detection: 4 BCs and 4 HCs	miR-146a	↑	-	-	-	Diagnosis: low-grade NMIBC vs. HC
	Validation: 27 NMIBCs, 7 MIBCs, and 9 HCs	miR-375	↓	-	-	-	Diagnosis: high-grade NMIBC vs. HC
**Plasma/Serum**							
Yan et al. 2020 [56]	Cohort 1: 3 BCs and 3 HCs	miR-4644	↑	-	-	-	Diagnosis: BC vs. HC
				-	-	-	Prognosis: ↑ with tumor stage
	Cohort 2: 25 NMIBCs, 32 MIBCs, and 25 HCs	miR-4298	↓	-	-	-	Diagnosis: BC vs. HC
		miR-4669					
Yin et al. 2019 [57]	3 NMIBCs, 60 MIBCs, and 59 HCs	miR-663b	↑	-	-	-	Diagnosis: BC vs. HC

Abbreviations: AUC—area under the curve; BC—bladder cancer; EAU—European Association of Urology; HC—healthy control; NMIBC—non-muscle-invasive bladder cancer; MIBC—muscle-invasive bladder cancer; miRNA—microRNA; RFS—recurrence-free survival; PFS—progression-free survival; SN—sensitivity; SP—specificity; vs.— versus; ↑—higher; ↓—lower.

**Table 2 ijms-24-06757-t002:** lncRNAs with potential in the management of bladder cancer patients.

**Study**	**Cohort**	**lncRNA**	**Levels**	**SN (%)**	**SP (%)**	**AUC**	**Clinical Significance**	
**Urine**								
Chen et al. 2022 [58]	Cohort 1: 4 BCs and 3 HCs	TERC	↑	78.65	77.78	0.836	Diagnosis: BC vs. HC	
	Validation: 105 NMIBCs, 23 MIBCs, 46 benign lesions, and 94 HCs							
Abbastabar et al. 2019 [59]	20 NMIBCs, 10 MIBCs, and 10 HCs	ANRIL	↑	46.67	87.5	0.7229	Diagnosis: BC vs. HC	
		PCAT-1		43.33	87.5	0.7292		
Zhan et al. 2018 [60]	Screening: 61 NMIBCs, 43 MIBCs, and 104 HCs	Combined panel: MALAT1, PCAT-1, and SPRY4-IT1	↑	62.5	85.0	0.813	Diagnosis: BC vs. HC	
		PCAT-1	↑	-	-	-	Prognosis: ↓ RFS in NMIBC	
	Validation: 50 NIMBCs, 30 MIBCs, and 80 HCs							
		MALAT1	↑	-	-	-		
**Plasma/Serum**								
Zhang et al. 2019 [61]	Training: 56 NMIBCs, 44 MIBCs, and 100 HCs	Combined panel: PCAT-1, SNHG16, and UBC1	↑	80.0	75.0	0.826	Diagnosis: BC vs. HC	
	Validation: 84 NMIBCs, 76 MIBCs, and 160 HCs	UBC1	↑	-	-	-	Prognosis: ↓ RFS	
Wang et al. 2018 [62]	52 BCs, 52 benign urologic diseases, and 52 HCs	H19	↑	74.07	78.08	0.851	Diagnosis: BC vs. HC and benign urologic diseases	
				-	-	-	Prognosis: ↑ survival	
Zheng et al. 2018 [63]	41 NMIBCs, 9 MIBCs, and 50 HCs	PTENP1	↓	65.4	84.2	0.743	Diagnosis: BC vs. HC	
Xue et al. 2017 [64]	15 NMIBCs, 15 MIBCs, and 30 HCs	UCA1	↑	80.0	83.33	0.878	Diagnosis: BC vs. HC	

Abbreviations: AUC—area under the curve; BC—bladder cancer; HC—healthy control; lncRNA—long noncoding RNA; NMIBC—non-muscle-invasive bladder cancer; MIBC—muscle-invasive bladder cancer; RFS—recurrence-free survival; SN—sensitivity; SP—specificity; vs.— versus; ↑—higher; ↓—lower.

**Table 3 ijms-24-06757-t003:** Proteins with potential in the management of bladder cancer patients.

Study	Cohort	Protein	Levels	SN (%)	SP (%)	AUC	Clinical Significance
Urine							
Suh et al. 2022 [65]	Discovery: 19 NMIBCs, 5 MIBCs, and 12 HCs	Combined panel: Cofilin-1, ITIH2, and Afamin	↑	88.0	81.3	0.845	Diagnosis: BC vs. HC
	Validation: 75 NMIBCs, 20 MIBCs, and 25 HCs						
Lee et al. 2022 [66]	Discovery: 4 BCs pre- and postsurgery	a2M	↑	93.3	34.8	0.809	Diagnosis: BC vs. benign urologic diseases
	Validation: 57 NMIBCs, 2 MIBCs, and 22 benign urologic diseases						
Igami et al. 2022 [67]	Cohort 1: 9 BCs and 4 HCs	Combined panel: CEACAM1, CEACAM5, and CEACAM6	↑	81.82	97.87	0.907	Diagnosis: BC vs. HC andBenign urologic diseases
	Cohort 2: 31 BCs, 18 benign urologic diseases, and 29 HCs						
Tomiyama et al. 2021 [68]	Discovery: 3 NMIBCs, 4 MIBCs, and 4 HCs	HSP90	↑	82.5	70.0	0.813	Diagnosis: BC vs. HC
		SDC1		82.5	63.3	0.785	
	Validation: 20 NMIBCs, 20 MIBCs, and 30 HCs						
		MARCKS		65.0	80.0	0.772	
Lee J. et al. 2018 [69]	Cohort 1: 5 NMIBCs, 4 MIBCs, and 8 HCs	MUC1, CEACAM-5, EPS8L2, and Moesin	↑	-	-	-	Diagnosis: BC vs. HC
	Validation: 4 NMIBCs, 2 MIBCs, and 6 HCs						
Lin et al. 2016 [70]	70 BCs, 59 ureter or renal pelvis cancers, 17 UTIs, 25 PCas, and 20 HCs	H2B1K α1AT	↑	-	-	-	Diagnosis: BC vs. HC and other urologic diseases
				-	-	-	Prognosis: ↑ with grade and stage
Chen et al. 2012 [71]	Discovery: 9 BCs and 9 hernias	TACSTD2	↑	73.6	76.5	0.80	Diagnosis: BC vs. hernia
	Validation: 28 BCs, 12 hernias, 5 hematurias, and 3 UTIs						
Smalley et al. 2008 [72]	4 BCs and 5 HCs	Resistin, GTPase Nras, EPS8L1, EPS8L2, RAI3, Mucin 4, EHC4EH, and α subunit of GsGTP-binding protein	↑	-	-	-	Diagnosis: BC vs. HC

Abbreviations: A2M—alpha-2 macroglobulin; AFM—afamin; APOA1—apolipoprotein A-I; AUC—area under the curve; BC—bladder cancer; CD5L—CD5 antigen-like protein; CDC5L—cell division cycle 5-like protein; CEACAM—carcinoembryonic-antigen-related cell adhesion molecules; CFL1—cofilin-1; EPS8L2—Epidermal growth factor receptor kinase substrate 8-like protein 2; FGB—fibrinogen beta chain; HC—healthy control; ITIH2—inter-alpha-trypsin inhibitor heavy chain H2; NMIBC—non-muscle-invasive bladder cancer; MIBC—muscle-invasive bladder cancer; PCa—prostate cancer; TACSTD2—tumor-associated calcium signal transducer 2; SN—sensitivity; SP—specificity; UTI—urinary tract infection; vs.— versus; ↑—higher; ↓—lower.

**Table 4 ijms-24-06757-t004:** Other and/or combined biomarker studies with potential in the management of bladder cancer patients.

**Study**	**Cohort**	**Biomarker**	**Levels**	**SN (%)**	**SP (%)**	**AUC**	**Clinical Significance**
**Urine**							
mRNA							
Zhu et al. 2021 [73]	11 NMIBCs, 24 MIBCs, and 35 HCs	Combined panel: STARD3NL, RPLP0, SF3A1, DDX17, RPL19, AUP1, CIT, PWP1, SLC46A3, SNX27, BICD2, ARL4C, PNMA5, EIF3CL, PPP2R2A, MT-ATP8, COL1A1 ^Ⴕ^, CD248 ^Ⴕ^, and PCGF5 ^Ⴕ^	↑ ^Ⴕ^↓	-	-	0.898	Diagnosis: BC vs. HC
Yazarlou et al. 2018 [74]	59 BCs, 24 HCs, and 25 benign urologic diseases	MAGE-B4	↑	71.7	66.7	0.67	Diagnosis: BC vs. HC
Perez et al. 2014 [75]	Cohort 1: 5 BCs and 6 HCs	GALNT1, LASS2, ARHGEF39, and FOXO3	↑	-	-	-	Diagnosis: BC vs. HC
	Validation: 3 BCs and 3 HCs						
Combined biomarker							
Huang et al. 2021 [76]	Training: 9 NMIBCs, 1 MIBC, and 10 HCs	Combined panel: mRNA—KLHCC7B, CASP14, and PRSS1 lncRNA—MIR205HG and GAS5 ^Ⴕ^	↑ ^Ⴕ^↓	88.5	83.3	0.924	Diagnosis: BC vs. HC
				87.2	83.3	0.91	Diagnosis: MIBC vs. NMIBC
	Validation: 64 NMIBCs, 16 MIBCs, and 80 HCs	MIR205HG GAS5	↓	-	-	-	Prognosis: ↓ PFS
				-	-	-	
Amuran et al. 2020 [77]	43 NMIBCs, 16 MIBCs, and 34 HCs	Combined panel: EVs miRNA—miR-139-5p, miR-136-3p, and miR-19b1-5p Circulating Protein—ApeRef1, BC4, and CRK	↑	80.0	86.4	0.899	Diagnosis: BC vs. HC
				93.3	95.5	0.976	Diagnosis: low-risk patients vs. HC
Berrondo et al. 2016 [78]	Cohort 1: 8 MIBCs and 5 HCs	HOTAIR, HOX-AS-2, MALAT, SOX2, and OCT4	↑	-	-	-	Diagnosis: BC vs. HC
	Cohort 2: 8 MIBCs and 5 HCs	HYMA1, LINC00477, LOC100506688, and OTX2- AS1	↑	-	-	-	Diagnosis: BC vs. HC
	Cohort 3: 10 MIBCs and 7 HCs						
**Plasma**							
miRNA and piRNA							
Sabo et al. 2020 [79]	39 NMIBCs, 8 MIBCs, and 46 HCs	miR-126-3p	↑	-	-	-	Diagnosis: G3 tumors vs. HC
		miR-4508	↓	-	-	-	Diagnosis: MIBC vs. HC
				-	-	-	Diagnosis: ↑ according to EAU risk class
		piR-5936	↑	-	-	-	Diagnosis: ↑ according to EAU risk class
		miR-185-5p	↓	-	-	-	Prognosis: ↓ survival
		miR-106a-5p					
		miR-10b-5p	↑	-	-	-	
**Urine and Serum**							
Chen et al. 2018 [80]	**Urine** 18 BCs and 14 HCs	PRMT5	↑	-	-	-	Diagnosis: BC vs. HC
	**Serum** 23 NMIBCs, 48 MIBCs, and 36 HCs			-	-	-	

Abbreviations: AUC—area under the curve; BC—bladder cancer; EAU—European Association of Urology; HC—healthy control; NMIBC—non-muscle-invasive bladder cancer; MIBC—muscle-invasive bladder cancer; miRNA—microRNA; piRNA—piwi-interacting RNA; PFS—progression-free survival PCa—prostate cancer; SN—sensitivity; SP—specificity; vs—versus; ↑—higher; ↓—lower.

## Data Availability

No new data was created or analyzed in this study. Data sharing is not applicable to this article.

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
