# Peer review of "Extracellular Vesicles as Potential Bladder Cancer Biomarkers: Take It or Leave It?"

_ijms, 2023, doi:10.3390/ijms24076757_

Round 1
Reviewer 1 Report
Evaluation of the EVs caro in liquid biopsy has been an interesting issue in medicine. They present diagnostic and prognostic values in different pathological circumstances. This review article is well written and included most of the published articles. Table will be more informative for readers. Please double-check published articles. The article would require being revised significantly to improve the English. Some sentences are unclear for example “Although providing EV samples with adequate recovery, most UC protocols re” and so on.
Author Response
Evaluation of the EVs caro in liquid biopsy has been an interesting issue in medicine. They present diagnostic and prognostic values in different pathological circumstances. This review article is well written and included most of the published articles. Table will be more informative for readers. Please double-check published articles. The article would require being revised significantly to improve the English. Some sentences are unclear for example “Although providing EV samples with adequate recovery, most UC protocols re” and so on. Reply: We thank the Reviewer for his/her positive opinion on our work. Furthermore, the manuscript was revised to improve English language, as advised. Namely, the sentence pointed by the reviewer was altered: “Although UC provides EV samples with adequate recovery, most UC protocols…” (line: 356).
Reviewer 2 Report
The Authors explore the current knowledge on the potential of bladder cancer extracellular vesicle-derived biomarkers and present their role as a non-invasive tool to assist in the diagnosis and management of bladder cancer patients. This is an important topic for society because bladder cancer is the 10th most diagnosed cancer worldwide. The work is well thought out and well organized. Generally, I have not found significant limitations in this manuscript; conversely, I think that it has many strengths, such as originality and accurate presentation of the issue.
Some technical remarks:
- Line 18: Double ‘Correspondence’.
- Line 20: 10th
- Line 345: No spaces before [84].
- Line 182: No explanation for FFPE.
Author Response
The Authors explore the current knowledge on the potential of bladder cancer extracellular vesicle-derived biomarkers and present their role as a non-invasive tool to assist in the diagnosis and management of bladder cancer patients. This is an important topic for society because bladder cancer is the 10th most diagnosed cancer worldwide. The work is well thought out and well organized. Generally, I have not found significant limitations in this manuscript; conversely, I think that it has many strengths, such as originality and accurate presentation of the issue.
Some technical remarks:
- Line 18: Double ‘Correspondence’.
- Line 20: 10th
- Line 345: No spaces before [84].
- Line 182: No explanation for FFPE.
Reply: We thank the Reviewer for the most positive comments on our work and for calling our attention to these matters. The points made by the Reviewer were revised as requested.
Reviewer 3 Report
The authors made an effort to review the literature on EV's in bladder cancer. Indeed, liquid biopsies are an interesting topic of research. At the same time, despite the constant search for markers for bladder cancer for over 20 years, unfortunately, we do not have a specific marker.
The work is interesting however, there are still several shortcomings in this paper that should be revised.
First of all, the structure of the review should be rebuilt
The introduction is far too long, multi-threaded and therefore not very comprehensive - it should be shortened and rebuilt. The section discussing the types and mechanisms of EV's can be a separate paragraph with additional discussion of the pro-oncogenic features of EV's
The section titled discussion should also be divided - as a single paragraph is long and tedious - separating from it a part focused on e.g. current challenges and another discussing future perspectives would definitely improve the clarity of the work. For the completeness of the analysis of the topic, also the potential therapeutic application, eg targeted therapy, should be discussed.
Since there are so many abbreviations a table for their elucidation would suffice.
I also suggest the abbreviation for bladder cancer as BC or Bca, which are by far more common in the literature than the BlCa used here
Author Response
The authors made an effort to review the literature on EV's in bladder cancer. Indeed, liquid biopsies are an interesting topic of research. At the same time, despite the constant search for markers for bladder cancer for over 20 years, unfortunately, we do not have a specific marker.
The work is interesting however, there are still several shortcomings in this paper that should be revised.
First of all, the structure of the review should be rebuilt.
The introduction is far too long, multi-threaded and therefore not very comprehensive - it should be shortened and rebuilt. The section discussing the types and mechanisms of EV's can be a separate paragraph with additional discussion of the pro-oncogenic features of EV's
Reply: We thank the Reviewer for this suggestion, which allowed us to improve the manuscript. As advised, changes were made in the Introduction section to rebuild and facilitate the reading. For instance, the sentence: “Cancer biomarker refers to any biological observation – protein, metabolite, RNA, DNA or even an epigenetic alteration - that can ideally replace and predict a clinically relevant outcome or an intermediate result that is more difficult to observe [25,26]” was removed. Moreover, subheadings, such as “1.1. Bladder Cancer” (line: 33) and “1.1.1. Epidemiology and biology” (line: 34) were added to clarify the meaning. Regarding the EVs’ mechanisms of action, the following paragraph was added to the “1.2.2. EVs’ physiological and pathological role” section (line: 122): “For instance, EVs from BC cells internalized by macrophages promote their polarization into pro-tumoral macrophages, enhancing the release of immunosuppressive cytokines, which facilitates tumor progression [46]. Additionally, transfer of EVs’ lncRNAs and miRNAs from cancer-associated fibroblasts to BC cells, showed chemotherapy resistance modulation [47,48]. Moreover, proteins derived from EVs of BC cells increased angiogenesis and migration of BC cells as well as endothelial cells, also facilitating cancer progression [49,50]. These studies provide evidence of EVs and its cargo’s pro-tumoral influence in BC.” (lines: 120-127).
The section titled discussion should also be divided - as a single paragraph is long and tedious - separating from it a part focused on e.g. current challenges and another discussing future perspectives would definitely improve the clarity of the work. For the completeness of the analysis of the topic, also the potential therapeutic application, eg targeted therapy, should be discussed.
Reply: Once again we thank the Reviewer for this suggestion, which really improved our point and the flow of our discussion section. Headings such as “4.1. The potential of EVs as BlCa biomarkers” (line: 307) and “4.2.1. Study design constraints” (line: 431) were added, as advised to clarify the message. Moreover, a “4.4. Future perspectives” (line: 382) was added with the respective text: “Considering the drawbacks of available biomarkers for BC detection and prognostication, novel biomarkers are urgently needed. Indeed, EVs’ derived cargo has been shown to be powerful tools as BC biomarkers. Additionally, several reports have already suggested EVs and their cargo as potential cancer vaccines in different cancer models [92]. Because therapeutic options are very limited in BC, it is of utmost importance to unveil alternative therapies, eventually using EVs as a therapeutic option. Indeed, a clinical trial is ongoing focusing on Chimeric EVs’ vaccines administration to treat patients with recurrent or metastatic BC (NCT05559177). Although promising, EV-related methodological hurdles are considerable. In the next years, research must focus on addressing the technical shortcomings of EVs isolation, not only by developing and standardizing techniques that might be easily (and reproducibly) implemented in clinical practice, but also by using methods that allow for accurate EV-cargo quantification (e.g., ddPCR or large-scale validation of housekeeping molecules for RT-qPCR analysis). The clinical validity issues must be also solved by increasing patient cohorts as well as by performing multicenter validation, to ascertain the real value of EVs as BC biomarkers.” (lines: 387-402).
Since there are so many abbreviations a table for their elucidation would suffice.
Reply: We thank the Reviewer for the suggestion that improved the clarity of our manuscript. Therefore, before the References section (line: 428) a list of abbreviations was introduced.
I also suggest the abbreviation for bladder cancer as BC or Bca, which are by far more common in the literature than the BlCa used here.
Reply: We thank the Reviewer for his/her suggestion. Thus, as bladder cancer abbreviation we chose to use BC instead of BlCa.
Round 2
Reviewer 3 Report
Authors addressed all comments.
The revised version of the manuscript has gained more clarity, consistency and proper focus on the topic.